

# MetaBAT, an efficient tool for accurately reconstructing single genomes from complex microbial communities

Dongwan D. Kang[1,2], Jeff Froula[1,2], Rob Egan[1,2] and Zhong Wang[1,2,3]

[1] Department of Energy Joint Genome Institute, Walnut Creek, CA, USA
[2] Genomics Division, Lawrence Berkeley National Laboratory, Berkeley, CA, USA
[3] School of Natural Sciences, University of California at Merced, Merced, CA, USA

## ABSTRACT

Grouping large genomic fragments assembled from shotgun metagenomic sequences to deconvolute complex microbial communities, or metagenome binning, enables the study of individual organisms and their interactions. Because of the complex nature of these communities, existing metagenome binning methods often miss a large number of microbial species. In addition, most of the tools are not scalable to large datasets. Here we introduce automated software called MetaBAT that integrates empirical probabilistic distances of genome abundance and tetranucleotide frequency for accurate metagenome binning. MetaBAT outperforms alternative methods in accuracy and computational efficiency on both synthetic and real metagenome datasets. It automatically forms hundreds of high quality genome bins on a very large assembly consisting millions of contigs in a matter of hours on a single node. MetaBAT is open source software and available at https://bitbucket.org/berkeleylab/metabat.

## INTRODUCTION

High throughput metagenome shotgun sequencing is a powerful tool for studying microbial communities directly taken from their environment, thereby avoiding the requirement of cultivation or the biases that may arise from it. Assembling short metagenome shotgun reads into larger genomic fragments (contigs) by short read assemblers (*Pevzner & Tang, 2001*; *Pevzner, Tang & Waterman, 2001*) often fails to produce full-length genomes. Predicting draft genomes from assembled metagenomic contigs by metagenome binning provides a substitute for full-length genomes (*Mande, Mohammed & Ghosh, 2012*; *Mavromatis et al., 2007*). Despite their fragmented nature, these draft genomes are often derived from individual species (or "population genomes" representing consensus sequences of different strains, (*Imelfort et al., 2014*), and they approximate full genomes as they can contain a near full set of genes.

Two metagenome binning approaches have been developed (reviewed in *Mande, Mohammed & Ghosh, 2012*). The supervised binning approach uses known genomes as references and relies on either sequence homology or sequence composition similarity

Corresponding author
Zhong Wang, zhongwang@lbl.gov

for binning (*Krause et al., 2008*; *Wu & Eisen, 2008*). This approach does not work well on environmental samples where many microbes do not have closely related species with known genomes. In contrast, the unsupervised approach relies on either discriminative sequence composition (*Teeling et al., 2004b*; *Yang et al., 2010*) or species (or genomic fragments) co-abundance (*Cotillard et al., 2013*; *Le Chatelier et al., 2013*; *Nielsen et al., 2014*; *Qin et al., 2012*; *Wu & Ye, 2011*) or both (*Albertsen et al., 2013*; *Alneberg et al., 2014*; *Imelfort et al., 2014*; *Sharon et al., 2013*; *Wrighton et al., 2012*; *Wu et al., 2014*) for binning. Recent studies have shown that species co-abundance feature can be very effective to deconvolute complex communities if there are many samples available (*Albertsen et al., 2013*; *Alneberg et al., 2014*; *Cotillard et al., 2013*; *Imelfort et al., 2014*; *Karlsson et al., 2013*; *Le Chatelier et al., 2013*; *Nielsen et al., 2014*; *Sharon et al., 2013*). A few recent methods, particularly CONCOCT (*Alneberg et al., 2014*) and GroopM (*Imelfort et al., 2014*), are also fully automated binning procedures.

Many of the above tools do not scale well to large metagenomic datasets. In this study, we developed MetaBAT (Metagenome Binning with Abundance and Tetra-nucleotide frequencies) as an efficient, fully automated software tool that is capable of binning millions of contigs from thousands of samples. By using a novel statistical framework to combine tetra-nucleotide frequency (TNF) and contig abundance probabilities, we demonstrated that MetaBAT produces high quality genome bins.

## MATERIALS AND METHODS

### An overview of MetaBAT software and its probabilistic models

As a pre-requisite for binning, the user must create BAM files by aligning the reads of each sample separately to the assembled metagenome (Fig. 1 steps from 1 to 3). MetaBAT takes an assembly file (fasta format, required) and sorted bam files (one per sample, optional) as inputs. For each pair of contigs in a metagenome assembly, MetaBAT calculates their probabilistic distances based on tetranucleotide frequency (TNF) and abundance (i.e., mean base coverage), then the two distances are integrated into one composite distance. All the pairwise distances form a matrix, which then is supplied to a modified k-medoid clustering algorithm to bin contigs iteratively and exhaustively into genome bins (Fig. 1).

We use tetranucleotide frequency as sequence composition signatures as it has been previously shown that different microbial genomes have distinct TNF biases (*Mrazek, 2009*; *Pride et al., 2003*; *Saeed, Tang & Halgamuge, 2012*; *Teeling et al., 2004a*). To empirically derive a distance to discriminate TNFs of different genomes, we calculated the likelihood of inter- and intra-species Euclidean distance (*Deza, 2012*) by using 1,414 unique, complete genome references from NCBI (Fig. 2A). This empirically derived distance is termed Tetranucleotide frequency Distance Probability (TDP).

To evaluate the effect of contig sizes on inter-species distance, we obtained posterior probability distributions of inter-species distance with several fixed sizes and observed better inter-species separation as contig size increases (Fig. 2B). As contigs in real metagenome assemblies have various sizes, we then modeled TDP between contigs of

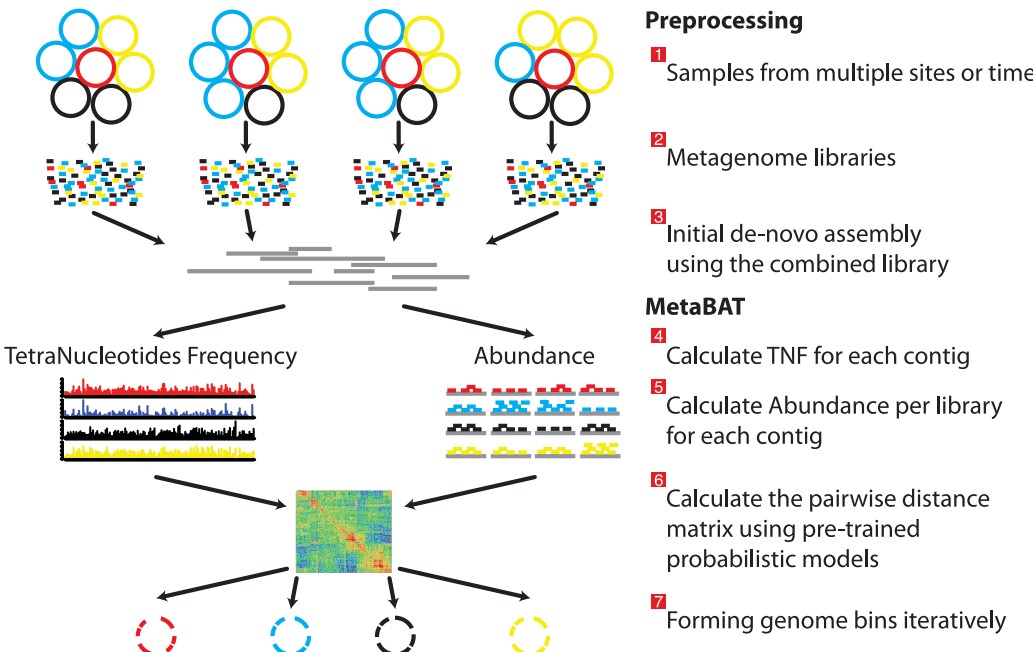

**Preprocessing**

1 Samples from multiple sites or times

2 Metagenome libraries

3 Initial de-novo assembly using the combined library

**MetaBAT**

4 Calculate TNF for each contig

5 Calculate Abundance per library for each contig

6 Calculate the pairwise distance matrix using pre-trained probabilistic models

7 Forming genome bins iteratively

TetraNucleotides Frequency

Abundance

**Figure 1 Overview of the MetaBAT pipeline.** There are three preprocessing steps before MetaBAT is applied: (1) A typical metagenome experiment may contain many spatial or time-series samples, each consisting of many different genomes (different color circles). (2) Each sample is sequenced by next-generation sequencing technology to form a sequencing library with many short reads. (3) The libraries may be combined before de novo assembly. After assembly, the reads from each sample must be aligned in separate BAM files. MetaBAT then automatically performs the remaining steps: (4) For each contig pair, a tetranucleotide frequency distance probability (TDP) is calculated from a distribution modelled from 1,414 reference genomes. (5) For each contig pair, an abundance distance probability (ADP) across all the samples is calculated. (6) The TDP and ADP of each contig pair are then combined, and the resulting distance for all pairs form a distance matrix. (7) Each bin will be formed iteratively and exhaustively from the distance matrix.

different sizes by fitting a logistic function to reflect the dynamic nature of the non-linear relationship between Euclidean TNF distance and TDP across different contig sizes. The results (Figs. 2C and 2D) suggest that the values of two parameters of the model, b and c are unstable if the size of either contig is very small (<2 kb) and one should be cautious to allow smaller contigs to be binned.

Although contigs originating from the same genome are expected to have similar sequence coverage, i.e., genome abundance, the coverage of contigs can vary significantly within a library due to biases originated from the current sequencing technology (*Benjamini & Speed, 2012*; *Harismendy et al., 2009*; *Nakamura et al., 2011*; *Ross et al., 2013*). As illustrated in Fig. 2E, the observed coverage variance derived from data consisting of isolate genome sequencing projects (total 99 from IMG Database (*Markowitz et al., 2012*), henceforth referred as the IMG dataset) significantly deviate from the theoretical Poisson distribution, consistent with the notion that both the variance and the mean should be modelled (*Clark et al., 2013*). For computational convenience, we chose the normal distribution as an approximation since it fits the observation much better
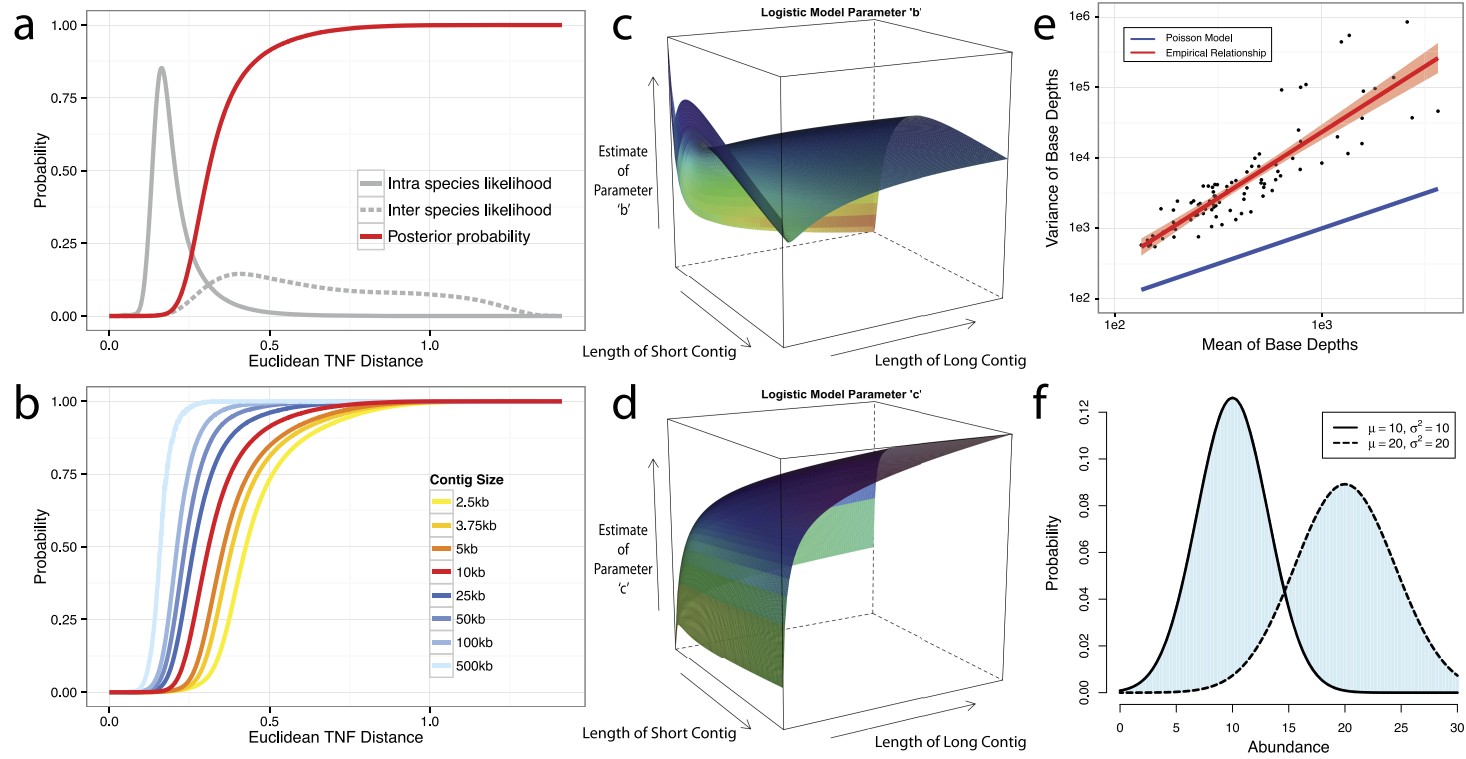

**Figure 2** **Probabilistic modeling of TNF and Abundance distances.** (A–D) TNF distance modeling. (A) Empirical probabilities of intra- (solid gray line) or inter- (dotted gray line) species Euclidean TNF distance are estimated from sequenced genomes. The posterior probability of two contigs originated from different genomes given a TNF distance is shown as a red solid line. All probabilities are calculated using a fixed contig size of 10 kb. (B) Different posterior inter-species probabilities for two equal-size contigs under various contig sizes. (C, D) The estimation of parameters for a logistic curve with two contigs of different sizes. $x$ and $y$ axis represent the lengths of short and long contig, respectively, and $z$ axis represents the estimates of each parameter $b$ or $c$ in a logistic curve, $\mathrm{TDP} = 1/(1 + \exp(-(b + c * \mathrm{TNF})))$, where TNF and TDP represents the Euclidean TNF distance and probabilistic TNF distance, respectively. (E–F) Abundance distance modeling. (E) The relationship between mean and variance of base depths (coverage) which were shown in $x$ and $y$ axis, respectively. Each dot represents this relationship in each genome, which calculated by median of mean and variance of the coverage. Theoretical Poisson model was shown as blue line and normal model was shown as red line. (F) Probabilistic abundance distance between two contigs. The shaded area represents the abundance distance between two contigs in a given library.

(Fig. 2E). To compute the abundance distance of two contigs in one sample, we use the area not shared by their inferred normal distributions with given coverage mean and variance (Fig. 2F). A geometric mean of the distances for all samples is used for the final abundance distance probability (ADP) of two contigs. In addition, we applied a progressive weighting mechanism to adjust the relative strength of the information from abundance distance, meaning that we put more weight on abundance distance when it was calculated from many samples (see below).

We then integrate TDP and ADP of each contig pair as the following:

$$P(\mu_1, \sigma_1^2, \mu_2, \sigma_2^2) = \begin{cases} \max(\mathrm{TDP}, \mathrm{ADP}), & \text{if } \mathrm{TDP} > 0.05 \\ \mathrm{ADP} \cdot w + \mathrm{TDP} \cdot (1 - w), & \text{otherwise} \end{cases}$$

where $w = \min[\log(n+1)/\log(m+1), \alpha] \cdot n, m$, and $\alpha$ represent the number of samples, a large number (100 as the default), and the maximum weight of ADP (0.9 as the default), respectively. For instance, in the default setting, the weight would be about 0.5 when the number of samples is 10 and TDP is less than 0.05. The resulting distance matrix is used for binning (see below).

## Tetranucleotide frequency probability distance (TDP)

To establish empirical probabilities of intra- and inter-species for tetranucleotide frequency distance, we downloaded 1,414 unique, completed bacterial genomes from the NCBI database and shredded them into fragments ranging from 2.5 kb to 500 kb. Next, we obtained 1 billion random contig pairs from within or between genomes. The empirical posterior probability that two contigs are from different genomes is given as the following:

$$P(T|D) = \frac{P(T)P(D|T)}{P(T)P(D|T) + P(R)P(D|R)}$$

where $T$ or $R$ represent cases where two contigs are from different (inter) or the same (intra) species, respectively. $D$ is the Euclidean TNF distance between two contigs. The same uninformative priors of $T$ and $R$ were chosen. In reality, $P(T)$ is expected to be much bigger than $P(R)$, thus we set $P(T) = 10 * P(R)$ as the default implementation to adjust for the possible under-sampling issue in inter species distance.

The TDP of contig pairs with different sizes is approximated using logistic regression:

$$P(D_{ij}; b_{ij}, c_{ij}) = \frac{1}{1 + e^{-(b_{ij} + c_{ij} * D_{ij})}}$$

where $D_{ij}$ represents a Euclidean TNF distance between contig $i$ and $j \cdot b$ and $c$, the two parameters for the logistic regression, are estimated from the empirical data.

## Abundance distance probability (ADP)

The probabilistic abundance distance was calculated as follows: Suppose two contigs have the mean coverage of $\mu_1$ and $\mu_2$ and the variances of $\sigma_1^2$ and $\sigma_2^2$, then we defined the abundance distance as the non-shared area of two normal distributions of $N(\mu_1, \sigma_1^2)$ and $N(\mu_2, \sigma_2^2)$:

$$P(\mu_1, \sigma_1^2, \mu_2, \sigma_2^2) = \frac{1}{2} \int |\phi_{\mu_1, \sigma_1^2} - \phi_{\mu_2, \sigma_2^2}|$$

where $\phi$ represents a normal distribution having two parameters $\mu$ and $\sigma^2$. Numerically this can be simplified using cumulative distribution functions as follows assuming $\sigma_2^2$ is greater than or equal to $\sigma_1^2$:

$$P(\mu_1, \sigma_1^2, \mu_2, \sigma_2^2) = \begin{cases} \Phi_{\mu_1, \sigma_1^2}(k_0) - \Phi_{\mu_2, \sigma_2^2}(k_0), & \text{if } \sigma_1^2 = \sigma_2^2 \\ \Phi_{\mu_1, \sigma_1^2}(k_2) - \Phi_{\mu_1, \sigma_1^2}(k_1) + \Phi_{\mu_2, \sigma_2^2}(k_1) - \Phi_{\mu_2, \sigma_2^2}(k_2), & \text{otherwise} \end{cases}$$
where $\Phi$ represents a cumulative normal distribution, and

$$k_0 = \frac{\mu_1 + \mu_2}{2}$$

$$k_1^* = \frac{\sqrt{\sigma_1^2 \cdot \sigma_2^2 \cdot ((\mu_1 - \mu_2)^2 - 2 \cdot (\sigma_1^2 - \sigma_2^2) \cdot \log(\sigma_2/\sigma_1))} - \mu_1 \cdot \sigma_2^2 + \mu_2 \cdot \sigma_1^2}{\sigma_1^2 - \sigma_2^2}$$

$$k_2^* = \frac{\sqrt{\sigma_1^2 \cdot \sigma_2^2 \cdot ((\mu_1 - \mu_2)^2 - 2 \cdot (\sigma_1^2 - \sigma_2^2) \cdot \log(\sigma_2/\sigma_1))} + \mu_1 \cdot \sigma_2^2 - \mu_2 \cdot \sigma_1^2}{\sigma_1^2 - \sigma_2^2}$$

$$k_1 = \min(k_1^*, k_2^*) \quad \text{and} \quad k_2 = \max(k_1^*, k_2^*).$$

To combine multiple abundance probabilities across different samples, we calculated the geometric mean of probabilities:

$$P_{ij} = \sum_n \mathbb{1}\{\mu_{in} > c \text{ OR } \mu_{jn} > c\} \sqrt{\prod_n P_{ijn}(\mu_{in}, \sigma_{in}^2, \mu_{jn}, \sigma_{jn}^2) * \mathbb{1}\{\mu_{in} > c \text{ OR } \mu_{jn} > c\}}$$

where $P_{ijn}$ represents the probability calculated from two abundances $\mu_{in}$ and $\mu_{jn}$, and c represents a cut-off for reasonable minimum abundance for a contig.

As metagenome assemblies contain many small contigs, whether or not to include them is a dilemma; including small contigs will likely improve the genome completeness, at a cost of genome quality because their larger abundance variations make it harder to bin them correctly. We tried to empirically determine a reasonable contig size cut-off by plotting the ratio of mean and variance from the IMG single genome dataset. Although most genome variances are much larger than their means, their ratio becomes stabilized after contig size increases to 2.5 kb (Fig. S1). Therefore, we used 2.5 kb or larger contigs for the initial binning. Smaller contigs can be recruited after the binning, based on their correlation to the bins (*Imelfort et al., 2014*).

## Iterative binning

We modified the k-medoid clustering algorithm (*Kaufman & Rousseeuw, 1987*) to eliminate the need to input a value for *k* and to reduce search space for efficient binning. Specifically, the binning algorithm works as follows:

1. Find a seed contig (e.g., having the greatest coverage), and set it as the initial medoid.
2. Recruit all other contigs within a cutoff distance (i.e., parameters p1 and p2) to the seed.
3. Find a new medoid out of all member contigs.
4. Repeat 2–3 until there are no further updates to the medoid. These contigs form a bin.
5. For the rest of the contigs, repeat 1–4 to form more bins until no contigs are left.
6. Keep large bins (e.g., >200 kb), and dissolve all other bins into free contigs.
7. (optional) For dataset with at least 10 samples, recruit additional free contigs to each bin based on their abundance correlation.
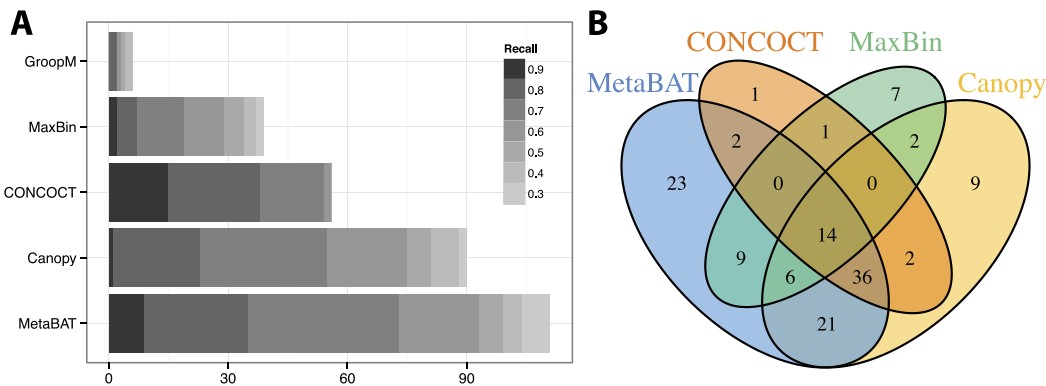

**Figure 3 Binning performance on synthetic metagenomic assemblies.** (A) The number of genomes (*X*-axis) identified by each binning method (*Y*-axis) in different recall (completeness) threshold and >90% precision, which calculates the lack of contamination. (B) Venn diagram of identified genomes by top 4 binning methods.

## RESULTS

### Binning performance on "error-free" metagenomic assemblies

A metagenomic dataset (Accession #: ERP000108) from the MetaHIT consortium (henceforth referred to as the MetaHIT dataset) (*Qin et al., 2010*) was chosen to benchmark MetaBAT because it contains a large number of samples and the community contains many species with reference genomes. To derive a reference genome set, we selected 290 known genomes from NCBI that are present in MetaHIT at >5× mean coverage (Table S1). These reference genomes were then shredded into contigs of random sizes (>2.5 kb) following an exponential distribution modeled to mimic real metagenome assemblies. The abundance of each contig in every sample was also obtained using real data. These "error-free" metagenome contigs, their abundance information, along with their parental reference genomes (as "true answers"), were used in the following analysis to benchmark binning performance. For a full description of the experiment, refer to the MetaBAT wiki page: https://bitbucket.org/berkeleylab/metabat/wiki/Home.

For comparison, we ran several alternative binning tools on the same dataset described above. These software include Canopy (*Nielsen et al., 2014*), CONCOCT v.0.4.0 (*Alneberg et al., 2014*), GroopM v.0.3.0 (*Imelfort et al., 2014*), and MaxBin v.1.4.1 (*Wu et al., 2014*). Among them, CONCOCT, GroopM, and MaxBin are also fully automated binning tools. An optional manual step in GroopM for improving the quality of bins was excluded. Since MaxBin does not consider multiple samples, we combined multiple samples into one.

We used >90% precision (lack of contamination) and >30% recall (completeness) as the minimum criteria for a bin to be considered "good" which basically means the bin should be composed of one or more strains of a single species (for results of other thresholds, refer to Figs. S3 and S4). Formulas for this calculation are described in the Supplemental Information 1. Among all binning tools, MetaBAT binned the greatest number of genomes at almost every recall threshold (Fig. 3A). CONCOCT is the only tool that produces more genome bins with over 80% or 90% completeness than MetaBAT,

**Table 1 Binning performance on synthetic metagenomic assemblies.**

|  | MetaBAT | Canopy | CONCOCT | MaxBin | GroopM[b] |
|---|---|---|---|---|---|
| Number of bins identified (>200 kb) | 340 | 230 | 235 | 260 | 445 |
| Number of genomes detected (Precision > .9 & Recall > .3) | 111 | 90 | 56 | 39 | 6 |
| Wall time (16 cores; 32 hyper-threads) | 00:13:55 | 00:21:01[a] | 104:58:01 | 20:51:19 | 45:29:39 |
| Peak memory usage (for binning step) | 3.9G | 1.82G[a] | 9.55G | 7.7G | 38G |

**Notes.**

[a] Canopy only use abundance table as input, so it should have taken more time and memory to read and write sequence data like the others.

[b] Manual steps were not used.

but MetaBAT produces many more bins at 70% completeness threshold. Interestingly, we found these tools complement each other in forming genome bins (Fig. 3B, GroopM was omitted since it detects only a few genomes). Among the unique 133 genome bins collectively formed by all tools, MetaBAT binned the most number of genomes (111, 83.5%), with 23 bins (17.2%) that were not found by any other tool.

MetaBAT runs very efficiently in computation; the entire binning process only took 14 min and 4 GB of RAM (Table 1). Multiple simulations produced almost identical performance results and thus were not shown.

## Binning performance on real metagenomic assemblies

We next tested the performance of MetaBAT on real metagenomic assemblies. Using the same raw sequence data from the above MetaHIT dataset, we pooled the sequences from all samples and then assembled them using Ray Meta assembler (*Boisvert et al., 2012*). Because real metagenomic assemblies often contain many small contigs, we lowered the minimum contig size requirement from 2.5 kb to 1.5 kb to include more contigs into our binning experiment. As a result, 118,025 contigs were used for binning. We then ran the above 5 binning tools with their default settings on this dataset. In contrast to the previous simulation experiment, in this experiment we do not have a reference genome for every genome bin; we instead used CheckM (*Parks et al., 2015*) to calculate the approximate recall (percent of expected single-copy-genes that are binned) and precision (the absence of genes from different genomes) rates. A full description of this experiment is available of MetaBAT wiki page: https://bitbucket.org/berkeleylab/metabat/wiki/Benchmark_MetaHIT.

Similar to the previous "error-free" experiment, MetaBAT again identified the greatest number of unique genome bins having >90% precision (Fig. 4A). In this experiment with real metagenomic contigs, the superior completeness we saw in CONCOCT during the "error-free" experiment was lost. Moreover, the number of genome bins formed by MetaBAT was consistently greater than the others at every completeness threshold. Similarly, different tools produced complementary binning results as before (Fig. 4B). MetaBAT's contribution appears to be more pronounced this time. It missed 17 bins formed by all other tools combined, but recovered 31 bins that no other tools produced. MetaBAT alone recovered 90.2% (133/144) of genome bins from all tools. These results suggest MetaBAT is very robust when run against a real metagenome assembly. Consistent

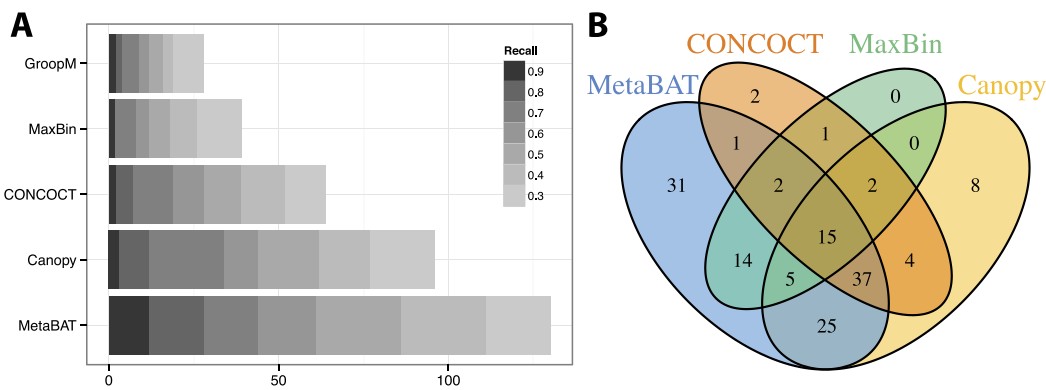

**Figure 4  Binning performance on real metagenomic assemblies.** (A) The number of genomes (*X*-axis) identified by each binning method (*Y*-axis) in different recall (completeness) threshold and >90% precision, which calculates the lack of contamination. (B) Venn diagram of identified genomes by top 4 binning methods.

**Table 2  A summary of the binning performance on real metagenomic assembly.**

|  | MetaBAT | Canopy | CONCOCT | MaxBin | GroopM[b] |
|---|---|---|---|---|---|
| Number of bins identified (>200 kb) | 234 | 223 | 260 | 168 | 335 |
| Number of genomes detected (Precision > 9 & Recall > .3) | 130 | 96 | 64 | 39 | 28 |
| Wall time (16 cores; 32 hyper-threads) | 00:03:36 | 00:02:31[a] | 82:19:53 | 06:49:39 | 12:19:12 |
| Peak memory usage (for binning step) | 3.0G | 1.6G[a] | 7G | 5.8G | 6.3G |

**Notes.**
[a] Canopy only use abundance table as input, so it should have taken more time and memory to read and write sequence data like the others.
[b] Manual steps were not used.

with the simulation experiment, MetaBAT is computationally very efficient and requires only 4 min to complete this experiment (Table 2).

To test the performance of MetaBAT on large-scale metagenomic data sets, we used a dataset containing 1,704 (with replicates) human gut microbiome samples (Accession #: ERP002061 (*Nielsen et al., 2014*). The entire dataset was assembled using Ray Meta assembler (*Boisvert et al., 2012*) and Megahit (*Li et al., 2015*) resulting in 1,058,952 contigs (>1 kb) that were then used for binning. MetaBAT took less than 2 h to generate 1,634 genome bins (>200 kb) using a single node with 16 CPU cores (32 hyper-threads), and the peak memory consumption was at 17G. In comparison, Canopy took 18 h using 36G memory in the same setting. The other binning tools—CONCOCT, GroopM, and MaxBin—all failed to generate any genome bins for this data set likely due to their inability to scale. For accuracy evaluation, we used CheckM (*Parks et al., 2015*) and identified 610 high quality bins (>90% precision and >50% completeness) among the bins predicted by MetaBAT, which is 35% more than the published CAG bins (*Nielsen et al., 2014*) and 11% more than Canopy bins using our assembly. For details on the use of MetaBAT on this large dataset please refer to: https://bitbucket.org/berkeleylab/metabat/wiki/Example_Large_Data.

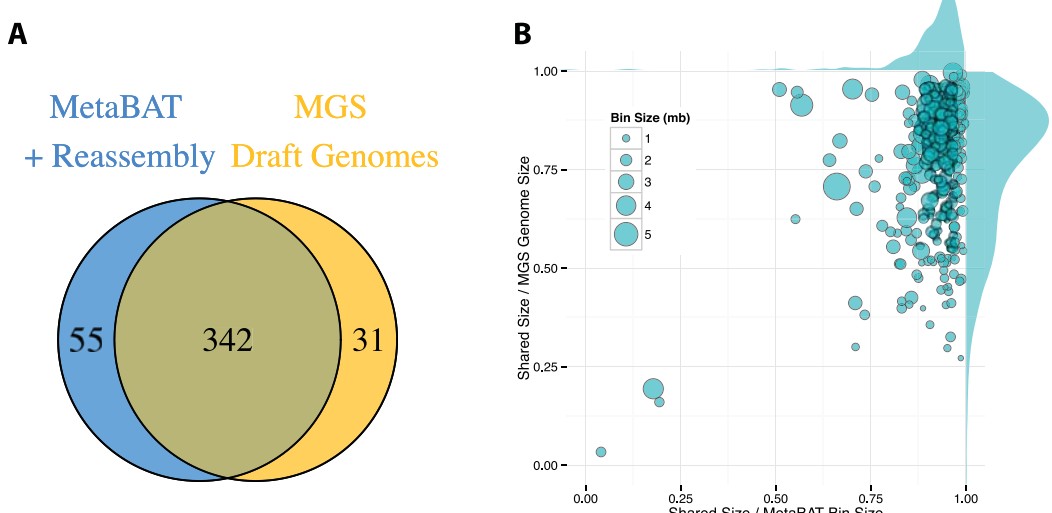

**A**

MetaBAT + Reassembly

MGS Draft Genomes

55   342   31

**B**

**Figure 5 Comparison between MetaBAT bins after post-processing and MGS draft genomes from Nielsen et al.** (A) Venn diagram of identified genome bins by MetaBAT having >90% precision and >30% completeness calculated by CheckM and one-to-one corresponding genomes in MGS draft genomes. (B) Scatterplot of completeness and precision for MetaBAT genome bins when considered MGS draft genomes as the gold standard. *X*-axis represents shared proportion of bases in terms of MetaBAT bins (i.e., precision), and *y*-axis represents shared proportion of bases in terms of MGS genomes (i.e., completeness). Each circle represents a unique MetaBAT bins having uniquely corresponding MGS genomes (342 bins in total), and the size of it corresponds to bin size.

## Post-binning processing to further improve quality

Assembly from pooled samples in the above experiment raises the possibility that similar genomes (e.g., different strains) present in different samples are assembled into chimeric contigs. This level of contamination may not be tolerated in some applications. Based on the assumption that a single sample will contain fewer strains of the same species than all pooled samples, we implemented an optional post-binning process to reduce the strain-level contamination. Briefly, we first selected a single sample with the most reads mapped to a specific bin, and then assembled these reads into a new set of contigs. If the new contigs produces better CheckM results, we subsequently replaced the old contigs in this bin with the new ones.

This post-processing step significantly improved both completeness and precision (lack of contamination) for 61% (992 out of 1,634) of the genome bins. Overall, there were 571 bins with >95% precision and >50% completeness, compared with 375 without post-processing. This improvement was more obvious when we increased the precision threshold to >99%, as the number high quality bins increases from 46 to 186.

By incorporating additional sequencing data and other post-binning optimizations, *Nielsen et al. (2014)* generated 373 high quality draft genomes ("MGS genomes"). We therefore used these MGS draft genomes as reference for additional quality assessment of the MetaBAT genome bins after post-processing. As shown in Fig. 5A, 31 MGS draft genomes were not well represented by MetaBAT bins, but MetaBAT recovered
55 additional genome bins not reported by MGS draft genomes. For those overlapping bins, most MetaBAT bins closely approximate the MGS draft genomes in accuracy—94% precision and 82% completeness (Fig. 5B).

## DISCUSSION

In conclusion, we developed an efficient and fully automated metagenome binning tool, MetaBAT, and evaluated its capability to reconstruct genomes using both synthetic and real world metagenome datasets. Applying MetaBAT to a large-scale complex human microbiome data recovers hundreds of high quality genome bins including many missed by alternative tools. An optional post-processing step improves the overall binning quality.

One limitation of this study is the choice of optimal parameters for binning. MetaBAT does not choose binning parameters automatically based on the underlying data. Instead, MetBAT provides five pre-set conditions that allows the user to select different levels of sensitivity and specificity (see online Software Manual for details). Users are strongly advised to explore the different presets to achieve the best result. Another limitation of this study is the choice of datasets. As the primary goal was to introduce a novel algorithm for metagenome binning, we only chose a synthetic dataset, a small real-world dataset and a large real-world dataset to test the performance of MetaBAT and compare it to alternatives. However, microbial communities can vary greatly in composition and structure. Similar comparisons applied to a different dataset might give different results. It is also advised for users to systematically evaluate different binners on a specific dataset for performance comparison, or to combine results from different tools for comprehensive binning.

Although binning methods evaluated in this study are all based on TNF, co-abundance, or both, the underlying algorithms are very different from one another. The algorithm implemented in MetaBAT is different from existing methods in several ways. First, MetaBAT uses different contig sizes to calculate posterior TNF probabilities. Second, MetaBAT dynamically weighs the TDP and ADP based on the number of samples. None of the existing tools adopt these two techniques. Finally, MetaBAT uses a scalable heuristics to iteratively cluster the contigs. To compute the pairwise contig distance matrix required for clustering, MetaBAT does not require a large number of samples as it uses the integration of TDP and ADP (when sample size is small, more weight is placed on TDP, see Methods). In contrast, the clustering algorithm employed in Canopy (*Nielsen et al., 2014*) does require a large number of samples as it is based on Pearson correlation coefficients.

One of the noticeable improvements in MetaBAT over other automated tools is its computational efficiency. In addition to the low memory requirement and fast computing speed, if one runs several rounds of binning to fine-tune parameters on a large dataset (by default MetaBAT does little parameter optimization), MetaBAT can be even faster as it saves intermediate calculations. For example, binning with ∼1M contigs and ∼1K samples for a second time only takes a few minutes.

There are a couple of considerations to keep in mind before applying MetaBAT. One important consideration is the minimum number of samples required for a reasonable binning performance. Although MetaBAT can run with only one sample or even in

TNF-only mode for binning, as shown in Fig. S6, our general advice is that more and diverse samples achieve better binning results. The greater the abundance variation among samples of a target species, the more likely MetaBAT will produce a good genome bin for this species. A second consideration is the quality of the metagenome assembly. We do not expect binning to work well with poor metagenome assemblies, e.g., assemblies including many small contigs less than 1kb, since the distance metrics computed for small contigs will not be very reliable.

## ACKNOWLEDGEMENTS

The authors thank Drs. Matt Blow, Rex Malmstrom and Tanja Woyke for their stimulating discussions and critical comments. The work was conducted by the US Department of Energy Joint Genome Institute.

### Funding

The work was supported by the Office of Science of the US Department of Energy under Contract No. DE-AC02-05CH11231. The funders had no role in study design, data collection and analysis, decision to publish, or preparation of the manuscript.

### Grant Disclosures

The following grant information was disclosed by the authors:
Office of Science of the US Department of Energy: DE-AC02-05CH11231.

### Competing Interests

The authors declare there are no competing interests.

### Author Contributions

- Dongwan D. Kang performed the experiments, analyzed the data, contributed reagents/materials/analysis tools, wrote the paper, prepared figures and/or tables, reviewed drafts of the paper.
- Jeff Froula performed the experiments, analyzed the data, wrote the paper, reviewed drafts of the paper.
- Rob Egan performed the experiments, analyzed the data, contributed reagents/materials/analysis tools, reviewed drafts of the paper.
- Zhong Wang conceived and designed the experiments, wrote the paper, prepared figures and/or tables, reviewed drafts of the paper.

### Supplemental Information

Supplemental information for this article can be found online at http://dx.doi.org/10.7717/peerj.1165#supplemental-information.

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
