# Peer review of "MetaBAT, an efficient tool for accurately reconstructing single genomes from complex microbial communities"

_PeerJ, doi:10.7717/peerj.1165_

## Round 0.1 · original submission · Major Revisions

Your manuscript has been seen by two knowledgable reviewers in the field. One of them has several questions that probably can be easily answered, the other reviewer would like to know more details about parameter choices, and I agree that this should be discussed in more detail. Also, the manuscript would require some assistance with language before it can be published.
Overall, the method and results are of high potential interest and we look forward to the revisions.

·

Basic reporting

The authors present a novel method for metagenome binning using a probabilistic distance based on the tetranucleotide frequency and coverage of contigs.
This pairwise distance between contigs is subsequently used in a medoid clustering algorithm to consolidate similar contigs into genome bins.
The paper is generally well-written and the ideas are clearly presented.

Experimental design

No comments

Validity of the findings

No comments

Additional comments

I have several minor remarks:
1) From the title and overview figure of the pipeline one would expect a tools that starts from the sequencing reads and outputs the genome bins.
It should be stated more clearly, that the assembled contigs and remapped reads to contigs are needed as input and Fig. 1 should indicate that this is not part of the tool.

2) In line 82 and 88 large number of samples are mentioned. Do these represent replicates or different samples?
More weight is put on the abundance distance if it is calculated from many samples, why is this the case?
If the sequencing depth is high enough, is it not sufficient to work even with one sample?

3) In section Tetranucleotide Frequency Probability Distribution (TDP):
Which genomic sequences were used from NCBI?
Were only prokayotic sequences used?
Is the approach applicable to prokaryotes only or also to microscopic eukaryotes?
The formula of the euclidean TNF distance should be shown.

4) Although this is out of the scope of the paper, it would be interesting to see how the resulting bins differ due to different sequencing platforms (with varying error rates and read lengths) and different assembly methods.

·

Basic reporting

The manuscript by Kang et al describes the design and evaluation of MetaBAT, a software tool designed to bin metagenomic assembly contigs into groups that represent genomes of species present in the sample(s). This topic has received much attention lately, since the first reports that combining information on contig abundance in different samples with sequence composition could be used to accurately group contigs from a single species (Albertsen et al 2013 Nat. Biotech.). The manuscript introduces a largely automated approach for assigning contigs to species groups, describing the principles behind the design of the algorithm, their implementation, and finally their evaluation on a human gut metagenome dataset.

On the whole the manuscript is structured in a way that is reasonable and logical. The description of related work in the introduction should really be expanded to describe earlier efforts to automatically bin based on coverage & composition. This would support a paragraph or two in the discussion section that gives the rationale for why those other methods/softwares could not simply be adapted or extended with the features described here. The approach does seem quite different from others I am familiar with, both in the theory and the implementation characteristics, but these differences could be better communicated.

The manuscript has a lot of minor grammar issues that really should be fixed to improve readability. There also seem to be some errors in the mathematical notation or at least opportunities for simplification which should be pursued in a revision.

In summary I think the manuscript needs revision but that this will ultimately be a valuable contribution to the field and I am looking forward to using the software in my own research in the future.

Experimental design

The approach taken is essentially ad-hoc. Equations are defined to calculate a distance between the tetramer nucleotide frequencies (TNF) in contigs of different lengths. These distances are then used to compute an empirical probability of the contigs deriving from the same species. This empirical probability equation is calibrated with existing isolate genome data and includes a term to adjust for contig length (because TNF estimates on short contigs have higher variance). A second empirical probability is based on the abundance profile of a contig pair across samples and includes terms to account for variation in depth of coverage within a genome. These two values are then combined in an ad hoc manner to arrive at a final score for whether a pair of contigs belong to the same species. These pairwise scores are then used in a clustering procedure with an algorithm inspired by the classic k-mediods method.

The overall approach seems reasonable but also has a large number of parameters that appear to have been manually trained/fit to data. The manuscript does not really describe how this was done or what datasets were used to optimize the selection of these parameters. As described below in my general comments, this is essential information for a number of reasons, including the interpretation of the validity of the benchmarking. It needs to be part of a revision.

As for the benchmarking, it is encouraging to see that some effort has been made to compare the various tools which combine abundance & composition information for genome binning. That said, I am concerned that the way the results are presented gives the impression that parameters may have been (inadvertently) selected in a way that shows MetaBAT in a favorable light. For example, GroopM produces a large number of bins, but these appear to get excluded by the chosen precision & recall criteria. It would be helpful to see the overall precision & recall of the unfiltered results of each program.

In terms of the method, more precision could be added to the description on how additional contigs are recruited to existing first-round bins. The manuscript simply says 'correlation' and the documentation says Pearson correlation. I haven't checked the code. I think it's worth pointing out that genomic sequence data is well known to be compositional data (normalization is intrinsic in the sequencing process) and so methods like Pearson's correlation can frequently give misleading results. Two good references on this issue in the context of genomic data analysis are:
Friedman & Alm 2012 PLoS Computational Biology http://journals.plos.org/ploscompbiol/article?id=10.1371/journal.pcbi.1002687
Lovell et al 2015 PLoS Computational Biology http://journals.plos.org/ploscompbiol/article?id=10.1371/journal.pcbi.1004075

I raise this issue only as a suggestion for future improvement, not as a required or even requested revision. Even if the existing approach produces some spurious correlations the benchmarking results should give a candid perspective that includes these. The effect of any spurious correlations may be negligible overall.

Validity of the findings

The main question regarding the validity of the findings relates to how parameters for the method were chosen. The benchmarking needs to be done on a dataset that is as independent as possible from the data used for parameter training. Ideally each method would be tested on many different datasets but doing that type of benchmarking is a huge undertaking which other groups are trying to solve (e.g. the CAMI challenge).

Additional comments

Line 24: "individual draft genomes" -> Imelfort et al introduced the term "population genomes" to describe these bins, and I suggest using that term since it more accurately describes the result. These bins capture a whole species across the samples, and within-species strain variation means the bins produced by methods such as yours represent a consensus of many genomes.

Line 28: "by a de bruijn graph based short read assembler" -> this should be stated more generally, e.g. assembly may be required, not necessarily using de bruijn graphs.

Line 30: "despite of lacking full contiguity" -> grammatical error

Line 32: "soil may contain 18,000 unique organisms" -> doesn't the term "organism" refer to an individual entity of some species? Despite the problems with the the bacterial species concept I think 'species' is more appropriate in this context.

Line 35: "Supervised binning approach takes advantage..." -> another grammar error. The manuscript has many. I am going to stop noting them but suggest a thorough and careful proofread by a native english speaker.

Line 41: This statement is incorrect. Methods such as GroopM use both abundance and composition information to bin. Given how similar the proposed method is to GroopM, it is essential that the relationship to this prior work be better described here in the introduction and throughout the manuscript. The same is true of CONCOCT by Alneberg et al.

Lines 73-77: "the observed coverage variance...significantly deviate from the theoretical Poisson distribution" coverage in Illumina sequencing is well known to co-vary with local GC content. The extent of the covariance depends on the library prep approach used. That covariance is not modeled here but might be worth mentioning.

Line 87: this equation essentially yields an ad-hoc score for a contig pair. Ad-hoc approaches are fine especially when fully model based methods are difficult to implement. The manuscript gives some helpful rationale for the structure of the equation, e.g. downweighting the TDP when the ADP information should be strong, but how are parameters chosen and has there been any effort to train these with machine learning methods? Also, were alternative equations explored before arriving at this one? This sort of information would be valuable for other tool developers to understand what ground has been covered already and where to go next.

Line 99: It is confusing to introduce two different variables for the same quantity, namely whether a contig paire have an intra- or inter-species relationship. Unless the authors are suggesting both T and R can be true simultaneously?

Line 105: are b_{i,j} and c_{i,j} actually dependent on i, j? It's not clear how this would be the case, and contradicts with the description given in the legend of Figure 2. Also, what are the actual values determined by the empirical fit to data?

Line 105: Given that almost all contigs are going to have different sizes, why does the manuscript bother to introduce the equation on line 99?

Line 102-103: Can the prior probability of two contigs being from the same or different species be calibrated based on 16S rDNA amplicon data for various environments?

Line 112: Is this requirement applied after quality trimming? If so, would be good to state so.

Line 112: Is a MAPQ threshold used? If not, are ambiguously-mapping reads handled somehow?

Line 114: "MetaBAT accepts sorted BAM..." this is very useful information but belong in a section on usage, not this theory section.

Line 117: The approach used to calculate coverage variance should be described either informally or formally. Is this the variance of the per-site coverage? How are reads mapped ambiguously to many sites handled?

Line 119: What is the term i here, and why is it discrete? Normal distributions are continuous, don't you want an integral to calculate there differences?

Lines 123-125: Why is this being derived instead of just using the Hellinger/Bhattacharya distance result? Perhaps there is a good reason, but if so it's not obvious. I will comment that use of the * character to indicate multiplication seems very out of place here.

Lines 131-141: The approach of ignoring small contigs during an initial round of coverage binning then later adding small contigs in to clusters defined by large contigs was introduced by Imelfort et al 2014 in GroopM and should be cited here.

Line 149: Isn't it just steps 2 & 3 that are repeated? e.g. find new contigs within the cutoff to the new medioid, then update the medioid?

Line 151: Is this minimum bin size a user-controllable parameter? I can imagine some viral genomes being much smaller than 200kbp.

Lines 170-172: Can you please give exact version numbers for these tools since software behavior can change substantially from one version to the next.

Line 175: Why 90%? Does recall change much if precision is set to, say, 50%? For some applications it may not be necessary to have such high precision. If it can be demonstrated that the reported recall patterns exist across a wide range of precisions it is a much stronger result than reporting patterns at a single arbitrarily chosen precision level.

Line 192: You've made an effort to tune MetaBAT's parameters for this dataset by lowering the min contig size, thereby improving its performance. Were similar efforts made to tune other methods? Or were default parameters used?

Line 192: This raises a more general question about the development of MetaBAT, which is what datasets were used to guide the design and testing of the algorithm prior to this benchmarking experiment? It's unfortunately very rare that manuscripts describing new tools in our field describe this process in any detail, and yet it often profoundly influences (biases) the algorithm design and parameter choices. For this reason it's crucial to understand the relationship between the datasets used during development and the datasets used for testing.

Line 234: I think it's misleading to talk about improving precision and recall here -- you have defined a process which explicitly maximizes these metrics on the dataset so they can no longer be interpreted as objective measures of performance (as precision and recall are typically defined to be). This is can be easily fixed by referring to the metrics by another name, e.g. completeness (as seems to have already been done in the following sentence).

Lines 225-237: It needs to be stated clearly here that this approach to improving the resolution among strains is only applicable when the conserved marker genes used by CheckM in the strains are different enough that they assemble into separate contigs. For closely related strains (which are likely to represent the majority of strains in a population) such genes will end up coassembling into the same contig(s).

Line 238: redundancy in this sentence?


Notes about the software:

1. the "static" binary is not statically linked, at least not to glibc, gomp, or pthreads. This might cause problems for some systems. The kernel ABI used is 2.6.18 which should be old enough to work on most systems. I did not try compiling from source but am pleased to see that it is available.

2. For impatient users, it might help to move the description of runMetaBat.sh near the top of the document.

3. In the perl scripts, I would suggest using /usr/bin/env perl for the interpreter since not all systems will have the preferred perl installation at /usr/bin. One or two lines of comments in each script about the purpose/input/output would be nice.

4. The driver script runMetaBat.sh creates a lockfile, and when the process is killed with Ctrl+C this lockfile persists. It would be helpful to catch signal 15 in the script and clean up the lock before exiting.

5. To test the software I ran it on an infant gut microbiome timeseries dataset I collected a few years ago. The dataset has 45 samples and a highly fragmented assembly (20k contigs, N50 2.3kbp) of a low complexity community (only 3 species, and 4-5 strains coming in above 5% abundance). Running metabat on this was incredibly easy to do, I simply handed the assembly and sorted .bam files to the wrapper script. After a few minutes this gave 7 bins. These bins seem relatively pure, although they are on the small side, likely due to most of the genome being in very small contigs. I have run GroopM on this dataset previously and achieved very good results, possibly even better than what MetaBAT was able to produce, but this is not an objective comparison because I don't have reference genomes for most of the strains in the sample. Bottom line, MetaBAT seemed to work reasonably well out of the box on an independent dataset.

---

## Round 0.2 · Minor Revisions

Both reviewers acknowledge that their remarks have been addressed and most questions answered -- yet some of the answers are not reflected in the manuscript or are maybe not communicated clearly enough. Please consider both reviewers' comments seriously, as we will not do a third round of revisions.

·

Basic reporting

No comments

Experimental design

The authors claim in their rebuttal letter that they cite the TNF distance they use in their revised manuscript, but I did not find it in the methods section. It would be of interest, because the tetranucleotide signature can be normalized in different ways.

Validity of the findings

No comments

Additional comments

The authors have sufficiently answered all my previous comments, except the one mentioned above. Therefore, I would recommend to accept the manuscript, once this minor point has been fixed.

·

Basic reporting

Nearly all of my comments have been addressed. There are just a few lingering important issues noted below.

Experimental design

no further comments

Validity of the findings

In their response, the authors have acknowledged the problem of parameter training and careful separation of training & benchmark data, however the manuscript has not been updated to address the issue. All of the reported accuracy measures need to be interpreted with extreme caution by readers. It has not (yet) been demonstrated that MetaBAT can outperform other methods on the wide range of microbial communities that are currently being sequenced by labs around the world. These limitations really need to be communicated to the readers, front & center, right next to the results in the manuscript.

Additional comments

The discussion now describes the features that distinguish MetaBAT from each of its predecessors but it is still not clear why existing software couldn't be extended. It's possible, maybe even likely that there are strong reasons for a fresh start over, e.g. MaxBin, but the rationale is not given here.

There are still grammatical issues in the text, though these are not a major impediment to understanding.

---

## Round 0.3 · accepted · Accept

There are still some editing and readability issues to be resolved, maybe the publication process in PeerJ might correct some of the issues.

It would be helpful to actually state / define the genome "binning" problem. What is the input, what is the desired output?

For example,
reviewed in (Mande et al. 2012))
should be
reviewed by Mande et al (2012)

Commas that belong to displayed formulas are often shown in the text below the formula.

The METHODS section describing the method is often mixed with results. This is because intermediate results are useful to motivate the method, but it actually makes it harder to understand the method quickly.

As it is, I consider the manuscript acceptable but not very readable for a wide readership.